# Divergent RNA viruses infecting sea lice, major ectoparasites of fish

Tianyi Chang[1¤], Brian P. V. Hunt[1,2,3], Junya Hirai[4], Curtis A. Suttle[1,2,3,5,6]*

**1** Department of Earth, Ocean and Atmospheric Sciences, University of British Columbia, Vancouver, Canada, **2** Institute for the Oceans and Fisheries, University of British Columbia, Vancouver, Canada, **3** Hakai Institute, Campbell River, Canada, **4** Atmosphere and Ocean Research Institute, The University of Tokyo, Kashiwa, Japan, **5** Department of Microbiology and Immunology, University of British Columbia, Vancouver, Canada, **6** Department of Botany, University of British Columbia, Vancouver, Canada

¤ Current address: Bigelow Laboratory for Ocean Sciences, Boothbay, Maine, United States of America
* suttle@science.ubc.ca

**Data Availability Statement:** Raw reads of marine copepods have been submitted to the Sequence Read Archive under accession number SAMN17831641. Sequences of novel viruses associated with Caligus clemensi can be accessed

## Abstract

Sea lice, the major ectoparasites of fish, have significant economic impacts on wild and farmed finfish, and have been implicated in the decline of wild salmon populations. As blood-feeding arthropods, sea lice may also be reservoirs for viruses infecting fish. However, except for two groups of negative-strand RNA viruses within the order *Mononegavirales*, nothing is known about viruses of sea lice. Here, we used transcriptomic data from three key species of sea lice (*Lepeophtheirus salmonis*, *Caligus clemensi*, and *Caligus rogercresseyi*) to identify 32 previously unknown RNA viruses. The viruses encompassed all the existing phyla of RNA viruses, with many placed in deeply branching lineages that likely represent new families and genera. Importantly, the presence of canonical virus-derived small interfering RNAs (viRNAs) indicates that most of these viruses infect sea lice, even though in some cases their closest classified relatives are only known to infect plants or fungi. We also identified both viRNAs and PIWI-interacting RNAs (piRNAs) from sequences of a bunya-like and two qin-like viruses in *C. rogercresseyi*. Our analyses showed that most of the viruses found in *C. rogercresseyi* occurred in multiple life stages, spanning from planktonic to parasitic stages. Phylogenetic analysis revealed that many of the viruses infecting sea lice were closely related to those that infect a wide array of eukaryotes with which arthropods associate, including fungi and parasitic tapeworms, implying that over evolutionary time there has been cross-phylum and cross-kingdom switching of viruses between arthropods and other eukaryotes. Overall, this study greatly expands our view of virus diversity in crustaceans, identifies viruses that infect and replicate in sea lice, and provides evidence that over evolutionary time, viruses have switched between arthropods and eukaryotic hosts in other phyla and kingdoms.

## Author summary

Sea lice are parasitic copepods and the major ectoparasites of fish. They have significant impacts on wild and farmed fish, and have been implicated in the decline of salmon

at GenBank under accession numbers MZ484458-MZ484479, OQ316414-OQ316416. These sequences as well as those used in the analyses of viRNAs can also be found at https://doi.org/10.6084/m9.figshare.22089500. The sequence alignments of viral RdRp proteins used to construct the phylogenetic trees are available at https://figshare.com/articles/dataset/Sequence_alignment_sealice_viruses/21391155. Figs 5 and 6, respectively, were generated using the sequence-alignment reports (BAM files) and data file ('sRNA_count_genomic_location.csv') showing the genomic position and the number of mapped viRNAs. The data for Figs 8 and 9 are in 'estimated_abundance_viral_groups.xlsx'. These files can be downloaded from https://doi.org/10.6084/m9.figshare.22273306.

**Funding:** This research was supported by a Discovery Grant from the Natural Science and Engineering Research Council (NSERC) to C.A.S., a scholarship from the Chinese Scholarship Council (CSC) (201606330071) to T.C., a fellowship and research grant to J.H. from the Japan Society for the Promotion of Science (20H03057), and overall support for sample collection and processing from the Tula Foundation to C.A.S and B.P.V.H. The funders had no role in study design, data collection and analysis, decision to publish, or preparation of the manuscript. T.C. received salary support from the NSERC Discovery grant and Tula Foundation grant to C.A.S.

**Competing interests:** The authors have declared that no competing interests exist.

populations; yet, viruses of sea lice are largely unknown. Here, we analyzed transcriptomes and small RNAs from three key species of sea lice and identified 32 previously unknown RNA viruses, many of which were actively replicating. Not only do these data greatly expand the known viral diversity in copepods, phylogenetic analysis provides evidence that over evolutionary time there has been extensive transmission of viruses between arthropods and other eukaryotes. Viruses replicating in sea lice included hypo-like and sobemo-like viruses, which previously were not known to infect arthropods. This study advances our view of the diversity and evolution of RNA viruses associated with sea lice, obtains genetic blueprints of viruses infecting sea lice, and provides approaches that may be further used to identify unknown viral pathogens in other ecologically and economically important crustaceans.

## Introduction

Sea lice are copepods in the family Caligidae, which have parasitic life stages that feed on the skin, mucus, and blood of fish [1,2]. They include members of the genus *Caligus*, which infect a wide range of marine fish, and *Lepeophtheirus salmonis*, which mainly infect salmonids [3]. Sea lice infestations have major impacts on wild and farmed fish. For example, Atlantic Salmon (*Salmo salar*) is one of the most widely cultivated marine fish (FAO, 2016), and losses due to sea lice are estimated to cost the industry more than 430 million US dollars annually [4]. Sea lice have also been implicated in the decline of wild salmon populations [5,6], and contributing to reduced productivity and growth [7,8]. Notably, as mechanical vectors, sea lice can transfer viruses among fish, including infectious haematopoietic necrosis virus (IHNV), and pose a risk for disease outbreaks in farmed and wild fish [9].

Despite the impact of sea lice on aquaculture and wild fish populations, only five viruses associated with sea lice have been described; all are negative-sense RNA viruses in the order *Mononegavirales*, and were discovered from transcriptomes of *Lepeophtheirus salmonis* and *Caligus rogercresseyi* [10–13]. The viruses in *Lepeophtheirus salmonis* were shown to be transmitted both vertically and horizontally [10,11]. Subsequent studies suppressed viral replication in *L. salmonis* using double-stranded viral RNAs [14], suggesting that sea lice generate virus-derived small interfering RNAs (viRNAs) in response to viral infections, as is known for some eukaryotes [15–17].

RNAi that counters viral infections has been reported in arthropods, including mosquitos, penaeid shrimp, fruit flies, silkworms, bees and their parasitic mites, as well as in some other eukaryotes [18–26]. Since viRNAs are only generated when viruses replicate in their hosts, viRNAs can be used to infer viral infection. Additionally, viRNAs differ among viruses infecting nematodes, fungi, and arthropods in terms of their length and 5' base composition; therefore, the type of host in which a virus is replicating can be determined from characteristics of the viRNAs [27].

To explore the viral diversity and identify potential viral pathogens of sea lice, we investigated transcriptomes and viRNAs from *Lepeophtheirus salmonis*, *Caligus clemensi* (for which only transcriptomes were analyzed), and *Caligus rogercresseyi*. Our analyses reveal previously unknown viruses from all the recognized phyla of RNA viruses. Phylogenetic analysis revealed that many of the viruses belonged to monophyletic groups that filled evolutionary gaps between classified families and genera, and that their RNA-dependent RNA polymerase (RdRps) encoding sequences shared low identities with the closest classified relatives, consistent with these viruses representing new taxonomic groups. Through the identification of

canonical viRNAs found in arthropods, we also show that sea lice are the hosts of divergent RNA viruses, and for many of the viruses, their closest classified relatives (members of the *Virgaviridae*, *Solemoviridae*, *Partitiviridae*, *Totiviridae*, and *Hypoviridae*) [28] infect plants or fungi. Besides greatly expanding the known viral diversity in crustaceans, by analyzing antiviral RNA interference (RNAi) we report a wide array of viruses that replicate in sea lice and thus may be agents of disease.

## Results

### The diversity and evolution of RNA viruses harbored by sea lice

To explore the diversity of RNA viruses associated with sea lice, we analyzed transcriptomic data from *Lepeophtheirus salmonis*, *Caligus clemensi* and *Caligus rogercresseyi* for virus-like sequences. For *C. clemensi* and *C. rogercresseyi* we *de novo* assembled transcriptomic data; whereas, for *L. salmonis* we examined assembled transcriptomic data from the Transcriptome Shotgun Assembly Sequence Database (TSA). Sequences larger than 200 bp were searched against the NCBI nr database, removing potential false-positives, and assigned into major taxonomic groups of RNA viruses (see Materials and Methods). Most of the viral sequences were from previously unknown viruses, but were related to major groups within the five phyla of RNA viruses. These groups included established orders and families, specifically, the *Picornavirales*, *Mononegavirales*, *Ghabrivirales* ('Toti-like'), *Bunyavirales*, *Narnaviridae*, *Solemoviridae*, *Tombusviridae*, *Qinviridae*, *Partitiviridae*, *Endornaviridae*, *Virgaviridae*, *Chuviridae*, and *Hypoviridae* (Fig 1).

To place the newly identified sea-lice-associated viruses into the current taxonomic and evolutionary framework of RNA-viruses, a maximum-likelihood phylogenetic tree was constructed for each major evolutionary group of RNA viruses; the trees included representatives from established and unclassified groups of viruses that were the closest relatives of the newly identified viruses. The trees were inferred based on predicted amino-acid sequences encoding RdRp, the most conserved gene of RNA viruses. This analysis revealed that the newly discovered viruses were related to members of a wide array of taxonomic groups (Figs 2–4 and S1 Table).

Phylogenetic analyses revealed that highly divergent lineages of sea-lice viruses and their unclassified relatives occurred within all the established phyla of RNA viruses (Fig 1), and filled evolutionary gaps between established orders and families. For example, in the *Lenarviricota*, a virus associated with *C. clemensi* formed a deeply branching lineage with unclassified arthropod-associated viruses (AAVs), which together were a sister to the clade encompassing the *Narnaviridae* (Fig 2). Similarly, in the phylum *Pisuviricota*, a virus associated with *C. clemensi* was anchored to the base of the clade comprising all known hypo-like viruses. Others in the phylum *Pisuviricota* included viruses that were most closely related to viruses formerly classified in the *Solemoviridae*, *Partitiviridae*, and *Picornavirales* (Fig 2). Moreover, similar patterns were observed across phyla; viruses associated with sea lice were typically clustered within distinct, well-supported phylogenetic groups, which in some cases have branch lengths that are consistent with those separating families (Figs 2–4). The diversity of viruses also extended across different life stages and species of sea lice, with most of the identified viruses associated with *C. rogercresseyi* occurring in its multiple life stages, and in many cases closely related viruses were identified from different species of sea lice (Figs 2 and 4).

### A wide variety of RNA viruses infect and replicate in sea lice

To identify viRNAs in sea lice, we recruited publicly available sRNAs from *C. rogercresseyi* and *L. salmonis* to all virus-like sequences (> 600 nt) identified in the two species. Canonical

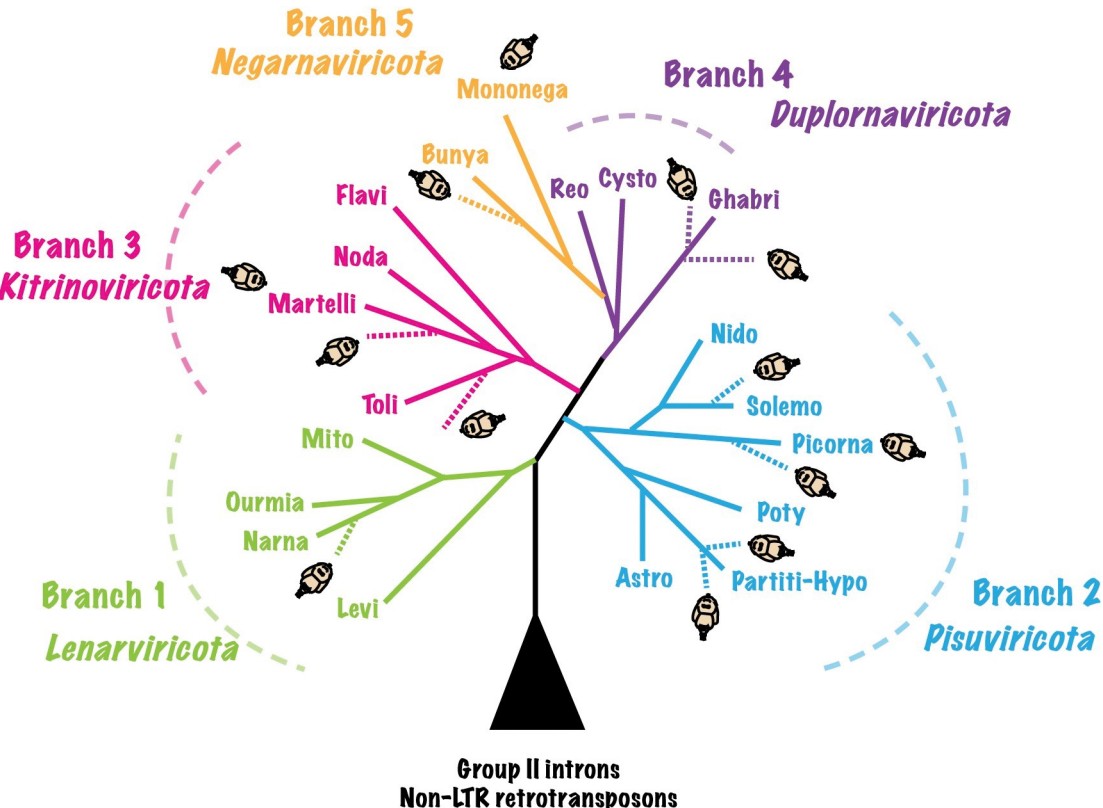

**Fig 1. A schematic of the major lineages of the newly identified sea-lice-associated RNA viruses in the tree of RNA viruses.**
The approximate phylogenetic positions of the sea-lice-associated RNA viruses are indicated by sea-lice icons and are based on analyses of the RdRp sequences for the viruses identified in this study. Levi, *Leviviricetes*; Narna, *Narnaviridae*; Ourmia, *Ourlivirales*; Mito, *Mitoviridae*; Partiti-Hypo, *Partitiviridae*, *Hypoviridae*, *Picobirnaviridae*, *Amalgaviridae*; Astro, *Astroviridae*; Poty, *Potyviridae*; Picorna, *Picornavirales*; Solemo, *Solemoviridae*; Nido, *Nidovirales*; Toli, *Tolivirales*; Martelli, *Martellivirales*; Noda, *Nodaviridae*; Flavi, *Flaviviridae*; Reo, *Reoviridae*; Cysto, *Cystoviridae*; Ghabri, *Ghabrivirales*; Bunya, *Bunyavirales*; Mononega, *Mononegavirales*. The colored branches indicate phyla recognized by the ICTV [28], and the phylogenetic relationships among them are based on Wolf et al. and Koonin et al. [29,30].

viRNAs similar to those found in other arthropods were generated from divergent RNA viruses, including hypo-like, picorna-like, virga-like, sobemo-like, toti-like, partiti-like, bunya-like, and qin-like viruses, as well as caligrhaviruses and hexartoviruses, indicative of viral replication.

Although the sequences indicated the viRNAs were derived from divergent viruses, most were 21-nt long with no strong strand or 5' base biases (Fig 5). These viRNAs were like those generated from RNA viruses infecting mosquitos and fruit flies and distinct from those produced by nematodes, plants, and fungi [33]. However, the viRNAs of the toti-like virus in *L. salmonis* were primarily derived from the positive strand, although a close relative of this virus, 'CARO Toti-like 1' found in *C. rogercresseyi*, gave rise to viRNAs from both strands more equally. Furthermore, the viRNAs were typically produced from the entire genomes of the targeted viruses (Fig 6). The PIWI-interacting small RNAs (piRNAs) produced from endogenous virus elements (EVEs) of *C. rogercresseyi* were predominantly 23–29 nt long, exhibited strong 5' nucleotide bias toward uridine, and were primarily generated from the negative strands of EVEs. We also observed both viRNAs and piRNAs from sequences of a bunya-like and two qin-like viruses in *C. rogercresseyi* (Fig 5).

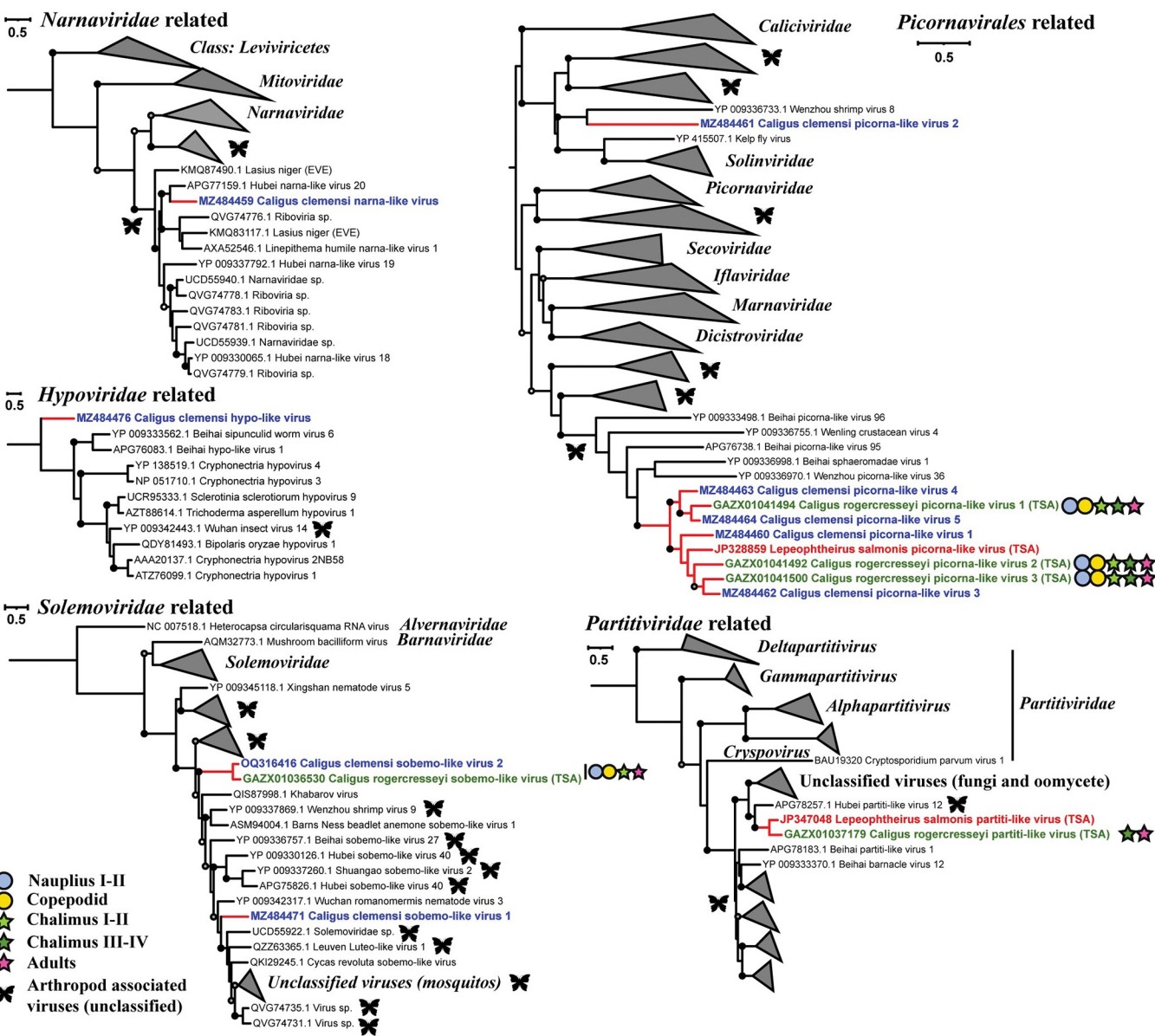

**Fig 2. Phylogenetic placement of sea-lice viruses for the phyla *Lenarviricota* and *Pisuviricota*.** Phylogenetic placement of sea-lice viruses within major taxonomic groups based on RdRp sequences for the phyla *Lenarviricota* (*Narnaviridae* related) and *Pisuviricota* (*Hypoviridae* related, *Solemoviridae* related, *Picornavirales* related, and *Partitiviridae* related). Branches representing members of established viral groups (i.e., families and genera) are collapsed, and branches representing unclassified arthropod-associated RNA viruses are collapsed and indicated with a butterfly icon. Viruses identified in the present study are indicated by red branches. Viruses discovered in different species of sea lice are color-coded by the host they are associated with: *Caligus clemensi* (blue), *Caligus rogercresseyi* (green), and *Lepeophtheirus salmonis* (red); those identified in different life stages of *C. rogercresseyi* are indicated by filled circles (planktonic stages) or stars (parasitic stages) at the end of each branch. Bootstrap branch support greater than 0.7, 0.8, and 0.9 are shown by empty, grey, and black circles, respectively. Each scale bar indicates 0.5 amino-acid substitutions per site. The maximum-likelihood trees are inferred based on the RdRp coding regions of the viruses with 1000 ultrafast bootstrap replicates using IQ-tree 2 [31]. The best substitution model is selected according to the BIC scores by ModelFinder [32] integrated within IQ-tree 2.

To provide further evidence that the viruses infect sea lice and not other sea-lice associated eukaryotes, we quantified the contigs derived from the transcriptomes of *C. clemensi*, then based on the taxonomy IDs assigned to the contigs (BLASTx against NCBI-nr database, see Materials and Methods), estimated the relative abundance (transcripts per million) for each taxon. Contigs

**Fig 3. Phylogenetic placement of sea-lice viruses for the phylum *Kitrinoviricota*.** Phylogenetic placement of sea-lice viruses within major taxonomic groups belonging to the phylum *Kitrinoviricota*. The figure legend is the same as for Fig 2, except the fish icons, which represent fish-associated viruses that are related to the viruses identified in sea lice.

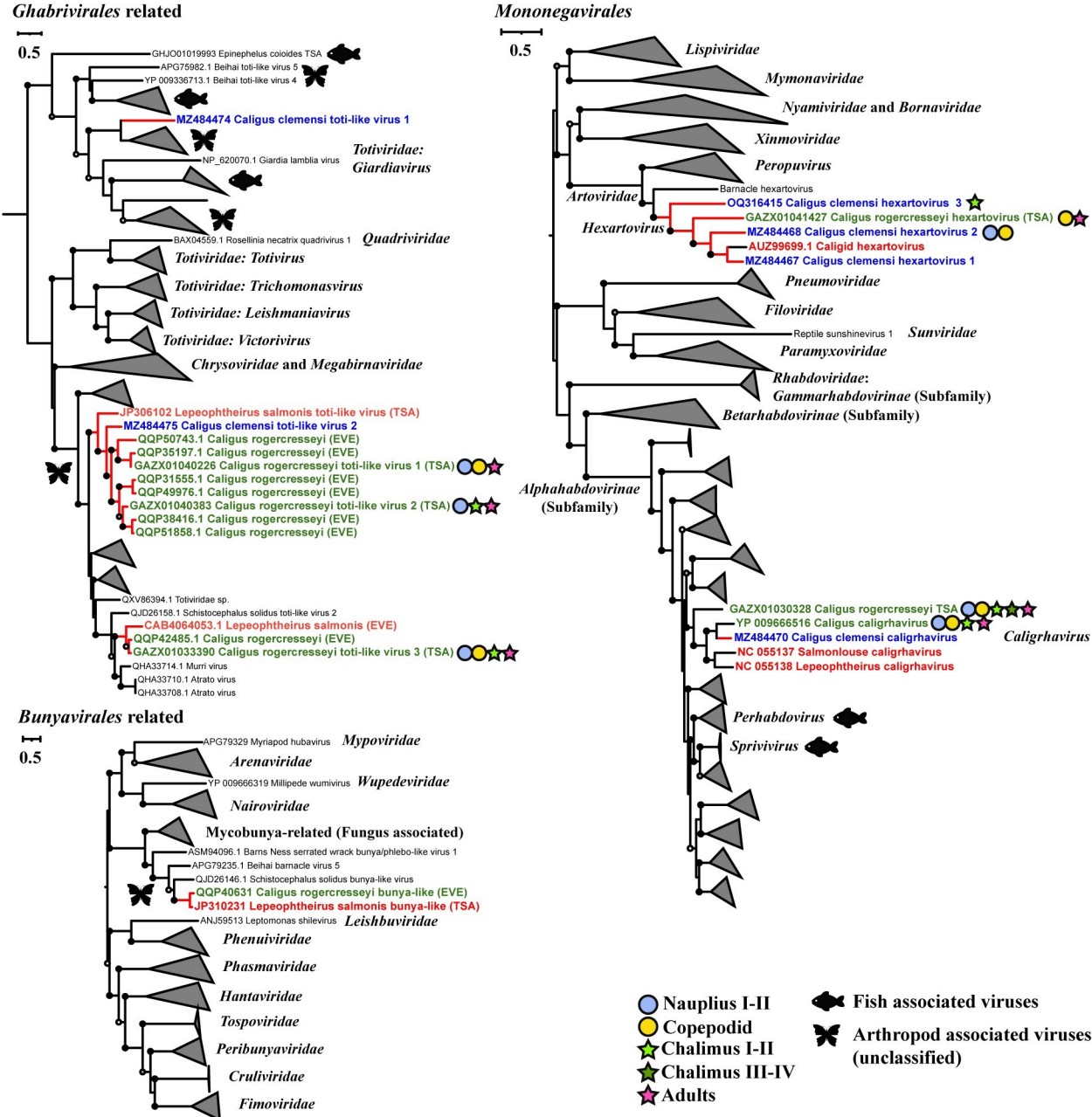

**Fig 4. Phylogenetic placement of sea-lice viruses for the phyla *Duplornaviricota* and *Negarnaviricota*.** Phylogenetic placement of sea-lice viruses within major taxonomic groups belonging to the phyla *Duplornaviricota* and *Negarnaviricota*. The figure legend is the same as that for Fig 2.

that showed significant sequence similarity to organisms that may be associated with sea lice, such as worms, fungi, algae and protists, accounted for less than 2% of the total abundance of classified contigs, which was even less than those assigned to viruses or bacteria (S1 Fig).

## Genomes of novel viruses that infect sea lice

We identified near-complete genomes for a wide array of previously unknown RNA viruses in sea lice (Fig 7). For instance, for the two sobemo-like viruses in *Caligus clemensi*, two genomic

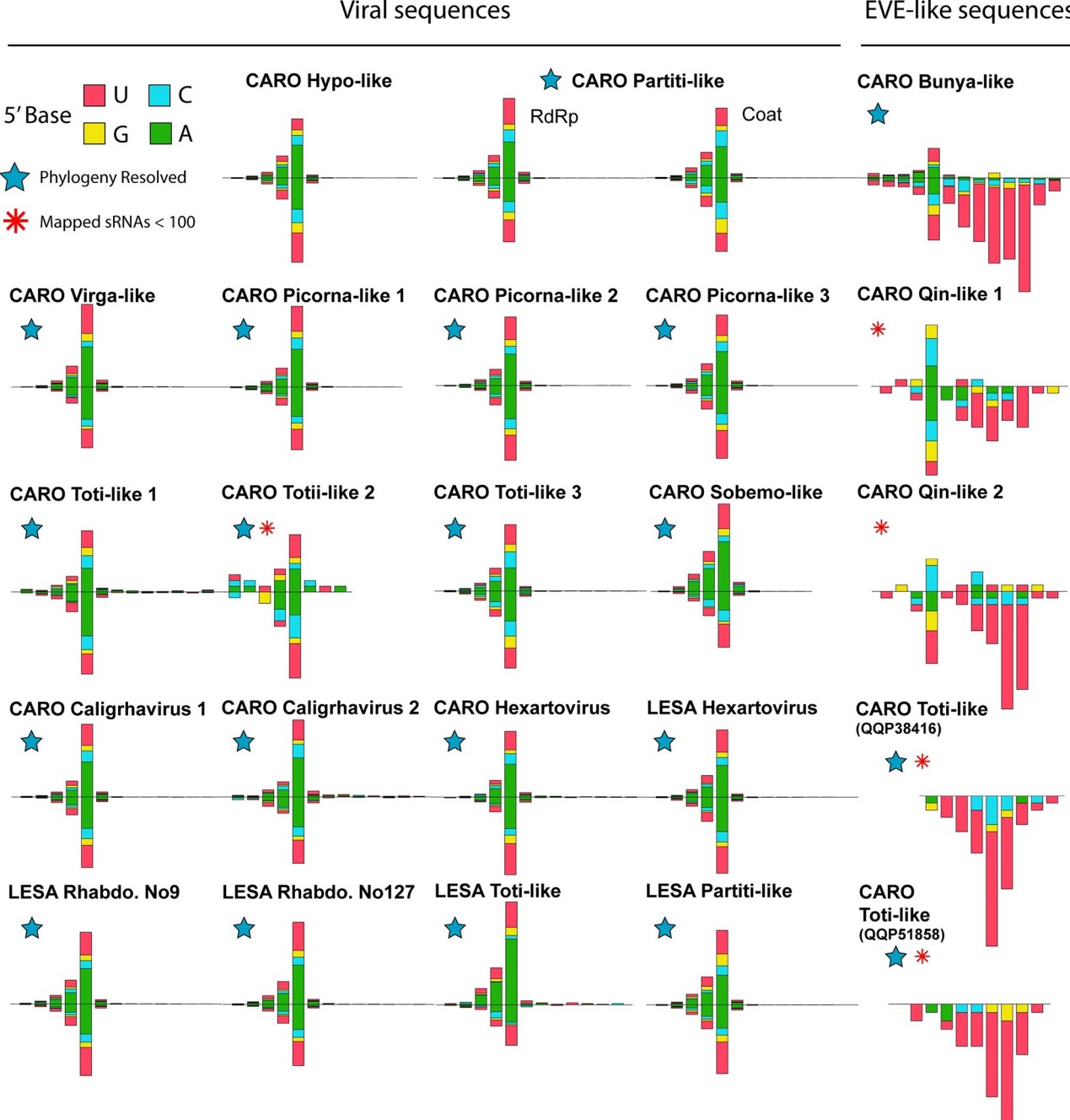

**Fig 5. Virus-derived small RNAs (viRNAs) and EVE-derived PIWI-interacting RNAs (piRNAs) found in *Caligus rogercresseyi* and *Lepeophtheirus salmonis*.** The small RNAs were mapped to sequences of RNA viruses discovered from the transcriptomes of *C. rogercresseyi*, in order to identify canonical viRNAs. Each bar plot represents the size distribution and 5' base composition of small RNAs that matched sequences of viruses discovered in *C. rogercresseyi*. The x-axis indicates the size of small RNAs with each axis beginning at 17 nt and ending at 29 nt. Bars above and below the x-axis indicate small RNAs mapping to the positive and negative senses of the virus, respectively. The y-axes, which are hidden to simplify the figure, indicate the number of aligned small RNAs for each size category (i.e., 17–29 nt). Bars are color-coded based on the frequency of 5-prime base of the mapped sRNAs (green: A, blue: C, yellow: G, pink: U). CARO, *Caligus rogercresseyi*. LESA, *Lepeophtheirus salmonis*. Blue stars indicate the viruses present in our phylogenetic trees. Red asterisks show the viruses that have fewer (< 100) detected sRNAs but display typical viRNA or piRNA patterns.

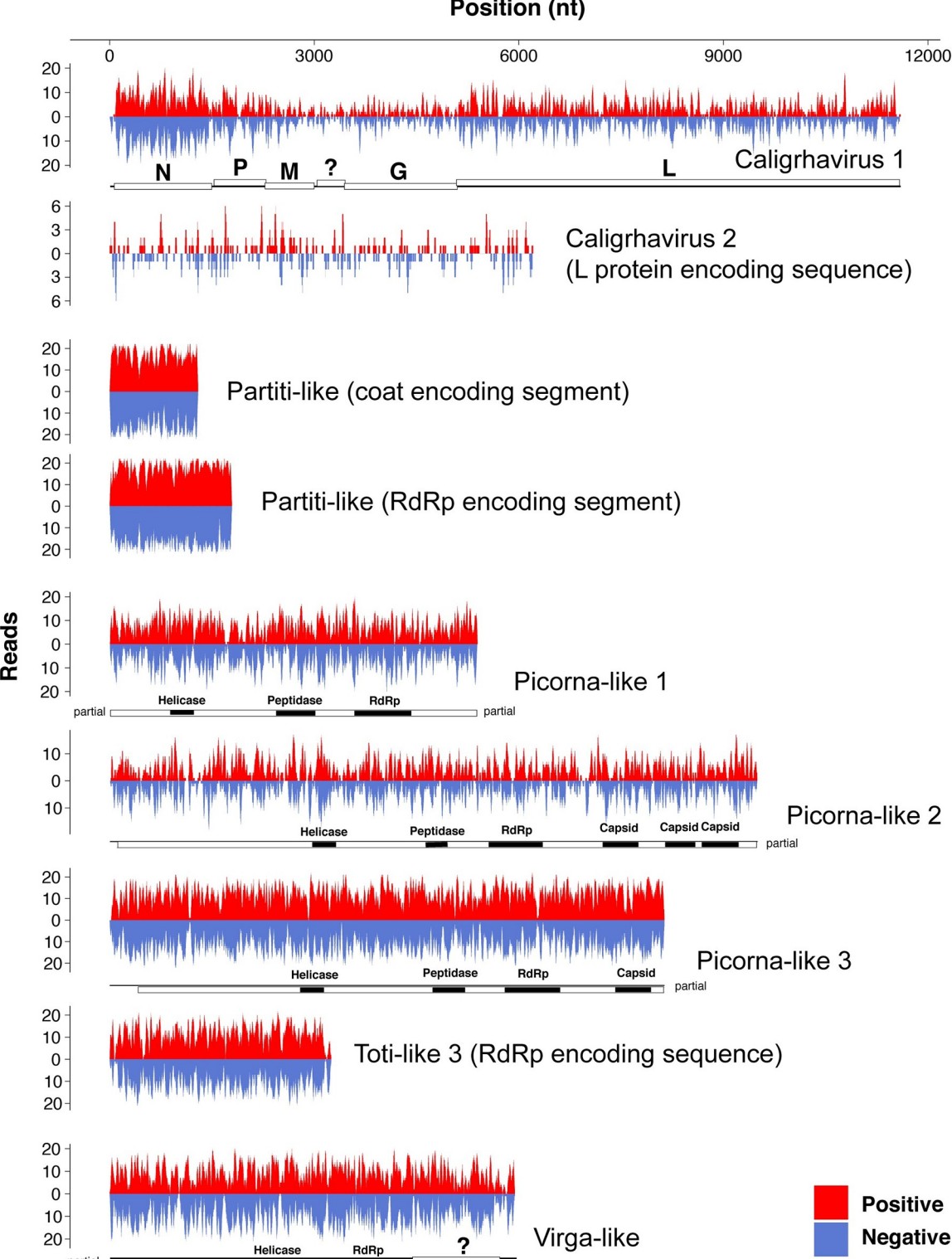

**Fig 6. The number and genomic location of viRNAs derived from the selected viruses in *C. rogercresseyi*.** Our analysis showed that viRNAs were typically generated from the entire viral sequences in *C. rogercresseyi*, while the number of the produced viRNAs varied markedly among genomic sites. N, nucleoprotein; P, phosphoprotein; M, matrix protein; G, glycoprotein; L, large protein that encompassing RdRp. The hypothetical protein is denoted by a question mark.

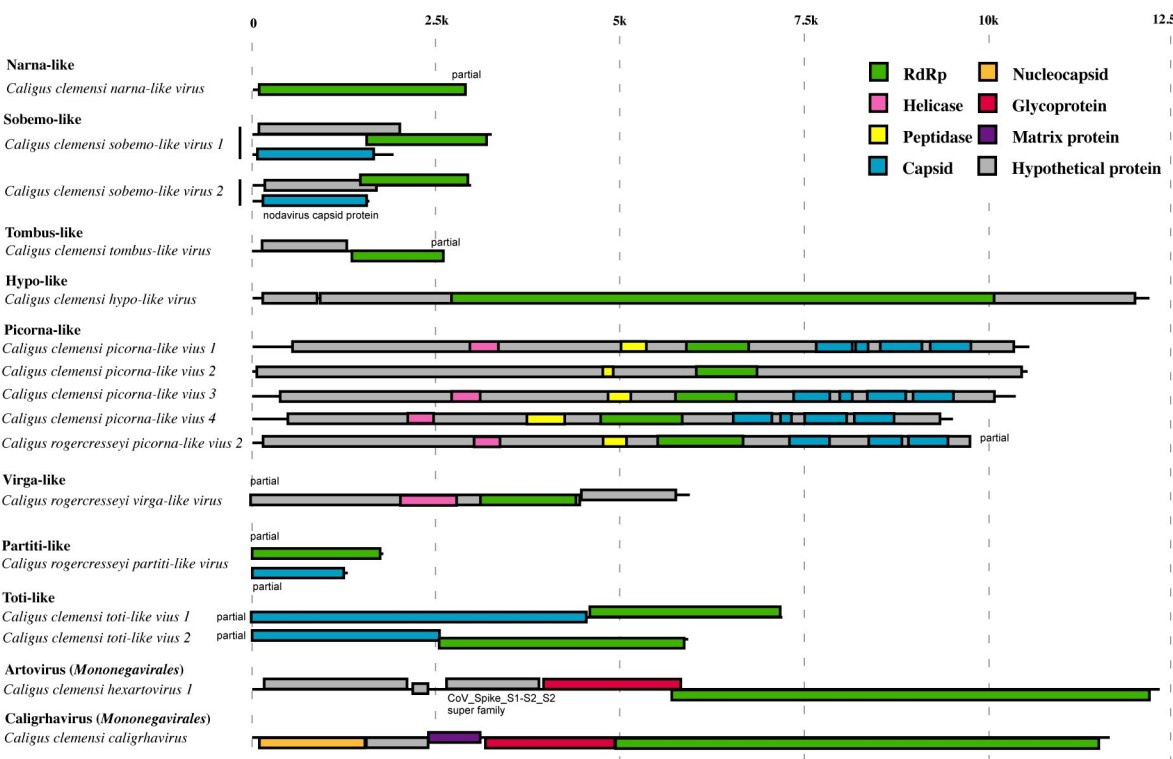

**Fig 7. Genome sizes and structures of previously unknown viruses found in sea lice.** The genome maps are arranged by phylogenetic group and by name within each group. Each viral genome is represented by a single line with the length of each line corresponding to genome size. The predicted open reading frames are shown as individual boxes on each genome map. Coding sequences are color-coded based on the functional genes they correspond to, while sequences encoding proteins with unknown functions are colored in grey.

segments were identified; the larger segment (3164 and 2886 nt for Caligus clemensi sobemo-like virus 1 and 2, respectively) comprised two overlapped ORFs, coding for a hypothetical protein and a RdRp, and the smaller segment (1886 and 1544 nt for Caligus clemensi sobemo-like virus 1 and 2, respectively) encoding a capsid protein. Similarly, segmented genomes were partially recovered for a partiti-like virus infecting *Caligus rogercresseyi*; the virus encoded genes for the capsid (>1297 nt) and RdRp (>1793 nt). Monopartite genomes were also characterized for narna-like, tombus-like, hypo-like, piconra-like, virga-like, and toti-like viruses, as well as two new members ('Caligus clemensi hexartovirus 1' and 'Caligus clemensi caligrha-virus') of the *Mononegavirales*. Consistent with their divergent RdRps, the size and gene content of these viruses were varied, ranging from a simple narna-like virus which encoded a single RdRp to more complex picorna-like viruses that encoded a helicase, peptidase, RdRp, and multiple capsid proteins.

In many cases, coding sequences of the newly discovered viruses were so divergent from those in the NCBI Conserved Domain Database that they could not be annotated by searching against the database (p value cutoff: 1e-2). Examples can be found for sobemo-, tombus-, hypo-, picorna-, and virga-like viruses, as well for the novel hexartovirus and caligrhavirus (Fig 7). There was also an unusual case where the capsid-encoding gene of the 'Caligus clem-ensi sobemo-like virus 2' shared significant sequence similarity to some noda- and permutote-tra-like viruses.

## Variation in the relative abundance of different types of viruses among sequencing libraries

For each sequencing library (accession numbers SRR15498757 to SRR15498762) [34] comprised of ten pooled individuals of *C. rogercresseyi*, collected at different life stages or from adult males or females, we estimated the abundance of viruses by taxonomic group (see https://doi.org/10.6084/m9.figshare.22273306). The number of viral sequences increased over two-fold from the non-feeding planktonic stages to the parasitic stages that live on fish (Fig 8). The relative abundance of viruses in each taxonomic group also varied among *C. rogercresseyi* collected at different life stages, as well as between female and male adults (Fig 9). Particularly,

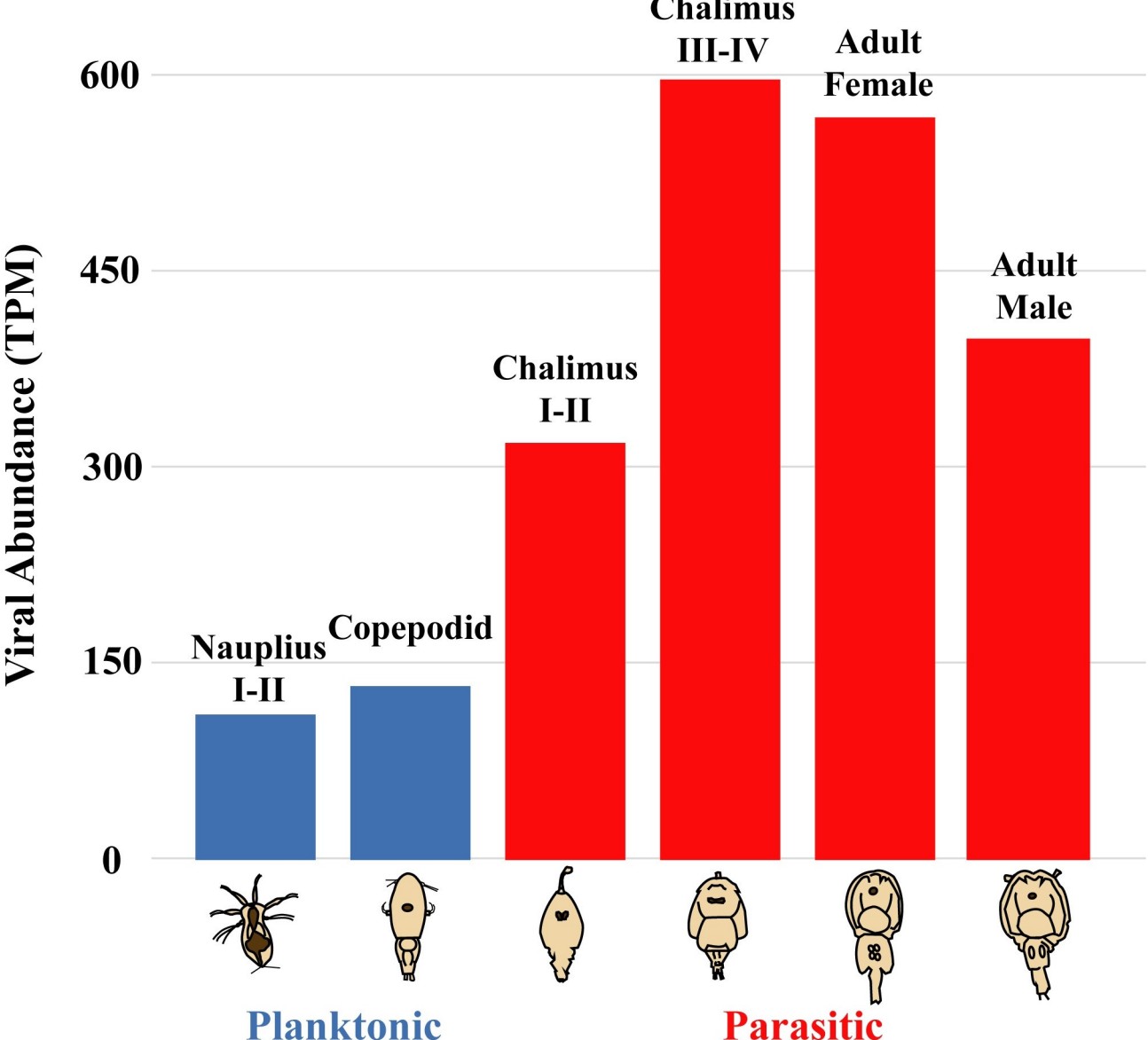

**Fig 8. Abundance of viral sequences in different life stages of sea lice (*C. rogercresseyi*).** The estimated abundance is based on polyA-tail enriched libraries; hence, includes viral genomes and transcripts with polyA-tails. Transcripts per million (TPM) are shown in parentheses for Nauplius I-II (110); Copepodid (133); Chalimus I-II (319); Chalimus III-IV (597); Female (567); Male (398).

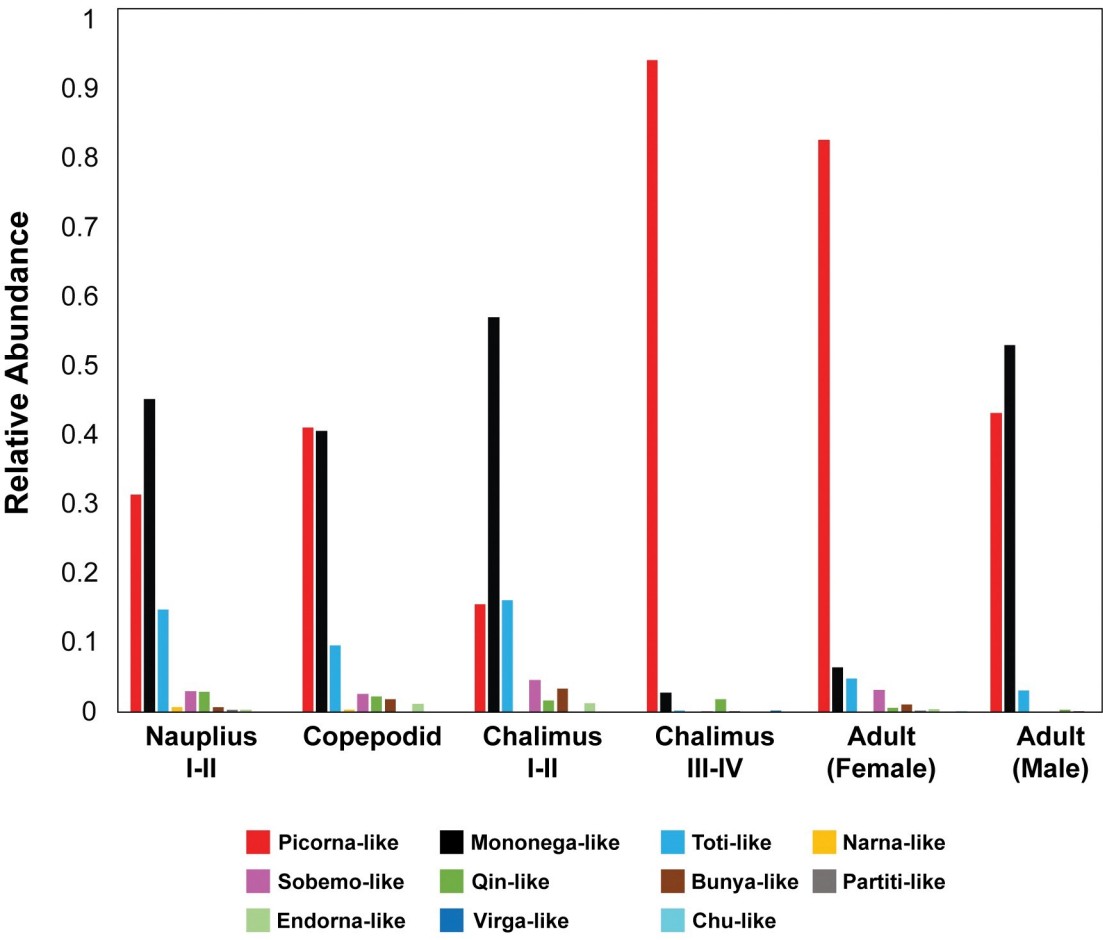

**Fig 9. Relative abundance of viruses in each taxonomic group in *C. rogercresseyi*.** Bars are color-coded according to the assigned taxonomic group. Relative abundance, estimated as the fraction of TPM for each taxonomic group, varied among *C. rogercresseyi* collected at different life stages and between males and females.

in the late-chalimus and female-adult stages, novel picorna-like viruses accounted for over 90% and 80%, respectively, of the estimated abundances of viruses. Among these newly identified viruses, 'Caligus rogercresseyi picorna-like virus 2' and 'Caligus rogercresseyi picorna-like virus 3' (Fig 2) were estimated to comprise about 80% of the viruses in chalimus III-IV and 39% in females of *C. rogercresseyi*, respectively.

## Discussion

Copepods are crustaceans, and among the most abundant animals on Earth [35]. They include parasitic species like sea lice, which are the major ectoparasites of fish, and a potential reservoir of fish pathogens [4]. Despite their ecological and economic importance, the diversity of viruses associated with copepods has remained largely unexplored. In this study, we investigated the sea-lice species *Caligus clemensi*, *Caligus rogercresseyi*, and *Lepeophtheirus salmonis*, and uncovered 32 previously unknown viruses that encompass all the established phyla of RNA viruses. Moreover, by analyzing virus-derived small RNAs (viRNAs) that are produced by host cells in response to infection, we showed that most of these viruses infect sea lice. We also found that both viRNAs and piRNAs were generated from some of the viruses, consistent

with sea lice possessing piRNA-based antiviral immunity, as occurs in the mosquito, *Aedes aegypti* [36]. Many of the newly identified viruses form deep-branching lineages with other unclassified viruses, consistent with being members of new families and genera. As well, many of these viruses have open reading frames (ORFs) with unrecognizable similarity to annotated genes, and possess novel genome arrangements, further emphasizing their distant relationship to extant viral taxa. Phylogenetic analyses also revealed numerous instances of horizontal transmissions of viruses between arthropods and distantly related eukaryotes. Below we elaborate on these findings and their significance.

## An expansion of the known virus diversity associated with copepods and its implications for viral taxonomy

Although many RNA viruses are known from arthropods [19,37–40], only five have been identified from copepods, all of which are negative-sense RNA viruses belonging to the order *Mononegavirales*. Several DNA viruses have also been found to be associated with crustacean zooplankton, including copepods [41,42] and freshwater crustaceans [43–45]; and it has been suggested that these viruses may contribute to short-term population declines of their associated hosts. Here, we show that previously unknown viruses related to those in the *Narnaviridae*, *Hypoviridae*, *Solemoviridae*, *Picornavirales*, *Partitiviridae*, *Tombusviridae*, *Virgaviridae*, *Ghabrivirales*, *Bunyavirales*, *Alphaendornavirus*, *Hexartovirus*, and *Caligrhavirus* are harbored by sea lice. These results greatly expand the known RNA-virus diversity associated with copepods, and crustaceans in general.

Our identification of novel deep-branching lineages across the five phyla of RNA viruses indicates that the viruses infecting sea lice represent divergent and previously unknown evolutionary groups. For example, the deepest-branching phylum of RNA-viruses is the *Lenarviricota* (Fig 1); although some narna-like viruses associated with *Lenarviricota* have recently been reported from arthropods [38,40,46], classified members of this phylum are only known to infect bacteria and fungi [28]. We identified a related virus from *C. clemensi*, 'Caligus clemensi narna-like virus', which falls within a lineage of unclassified AAVs and potential EVEs found in genomic sequences of the black garden ant (*Lasius niger*) (Fig 2). The closest described relative of this virus is 'Hubei narna-like virus 20', which was associated with a dipteran fly; however, sequence alignment only showed 41% average amino-acid identity (AAI) between the two viruses, indicating that they likely belong to separate genera. Furthermore, the lineage encompassing 'Caligus clemensi narna-like virus' is sister to the branch containing the *Narnaviridae* and another unclassified AAV lineage (Fig 2), suggesting that 'Caligus clemensi narna-like virus' and its closest relatives found in other arthropods are representatives of a new family in the *Lenarviricota*. Similarly, in the phylum *Pisuviricota*, despite almost all previously reported hypo-like viruses being fungi-associated, we unearthed a virus replicating (Fig 5) in sea lice that is at the base of known hypo-like viruses (Fig 2). The low AAI (< 24%) of RdRp-encoding sequences shared by 'Caligus clemensi hypo-like virus' and members of the *Hypoviridae* is consistent with this virus representing a new family. Additionally, deep-branching lineages formed by viruses reported here and their unclassified relatives have branch lengths that are consistent with those separating families for the groups 'Solemoviridae related', 'Picornavirales related', 'Virgaviridae related', 'Ghabrivirales related', and 'Bunyavirales related' (Figs 2–4). We therefore proposed to establish new taxonomic groups to accommodate these unassigned viruses.

## Identification of viral infections in sea lice through viRNAs and piRNAs

The presence of canonical arthropod-generated viRNAs indicates that diverse RNA viruses infect sea lice. These included hypo-like and sobemo-like viruses, for which, no relatives are

known to infect arthropods, although some have been detected in transcriptomes from arthropods [38,47–50]. Our analysis of viRNAs demonstrates that a diverse suite of RNA viruses infect and replicate in sea lice, and thus, may cause disease.

Notably, we observed both viRNAs and piRNAs from sequences of a bunya-like and two qin-like viruses in *C. rogercresseyi* (Fig 5). This implies that sea lice may use piRNA derived from EVEs (viral derived cDNA) as another RNAi-based antiviral immunity against viral propagation; such a mechanism has been shown in the mosquito *Aedes aegypti*, in which the Piwi4 protein has been found to be central in the piRNA-mediated immunity [36]. Indeed, homology searches using the sequence of *Aedes aegypti* Piwi4 (AAEL007698-PA, GenBank accession: EAT40579) as a query suggested that sea lice (GenBank accessions: QQP56097 and XP_040564360) also encode Piwi4-like proteins.

Importantly, we show here that viruses infecting copepods can be identified by sequencing viRNAs, and that viRNAs are derived from the full length of the viral genomes; thus, it should be possible to assemble complete viral genomes by sequencing and assembling copepod sRNAs directly, as has been done for other viruses [51]. As well, given that viRNAs accumulate during viral infection and display taxon-specific features, our results suggest that sequencing viRNA can be used to identify viruses infecting other copepods, including pelagic copepods.

In addition, this analysis provides genomic hotpots for sRNA production for diverse RNA viruses that infect sea lice, suggesting that these genomic locations are candidates for designing viral dsRNA for suppression of viral expression through the antiviral RNAi in sea lice [14].

## Sea lice are the hosts of the replicating viruses

Viral sequences in meta-omic data can only be used to indicate an association with a putative host, given that the sequences may have originated from free nucleotides, food, parasites, or symbionts. However, by analyzing viRNAs and EVEs, combined with phylogenetic analysis, we provide multiple lines of evidence of viral replication in sea lice, and demonstrate that sea lice are the bone fide hosts of the replicating viruses (viruses in Fig 5), as discussed below.

First, virus-derived small RNAs displayed highly conserved patterns, consistent with those generated from RNA viruses infecting insects and other members of the Pancrustacea [52]. These viRNAs are mostly 21–23 nt long, with no strong 5-prime base or strand preferences [19,25]. In contrast, viRNAs of fungus-infecting viruses predominantly have uridine at the 5-prime position [15,53,54], including a virus that infects an insect-associated fungus [53], and the Sclerotinia sclerotiorum hypovirus 2-L [54]. Whereas, the hypo-like virus we identified has the same canonical viRNA pattern as the other replicating viruses in sea lice, indicating that it is unlikely to replicate in a fungus. Nematodes and plants also generate particular viRNA patterns, with nematodes producing 22 and 26 nt small RNAs with a 5-prime bias for guanidine [16], while plants produce 21 or 22 nt viRNAs with a preference for 5' uridine or adenosine [17,55–57]. These distinct characteristics of viRNAs provide a basis for discerning the host of a replicating virus.

Second, by aligning virus-like sequences against genome assemblies of sea lice, we identified EVEs for toti-like and bunya-like viruses (Fig 4). EVEs are genetic "fossils" of recombination between a virus and a host, and provide strong evidence of a relatively long virus-host relationship [38]. Notably, the toti- and bunya-like viruses in sea lice are closely related to these EVEs, and form monophyletic groups. Moreover, small RNAs derived from the viruses and EVEs displayed canonical viRNAs and piRNAs, respectively, both are consistent with previous reports in other arthropods [58].

Third, less than 2% of the transcripts in *C. clemensi* were assigned to other eukaryotes, providing further evidence that eukaryotes other than sea lice, were unlikely to be hosts for the viruses. In summary, the evidence strongly supports sea lice being the hosts for the discovered viruses.

## Evidence for virus switching between arthropods and other eukaryotes

Many of the viruses that we discovered showed evidence of cross-phylum and cross-kingdom switching between arthropods and a wide array of eukaryotes with which arthropods associate. Viruses infecting arthropods have been postulated to be involved in the origin of plant [59] and vertebrate viruses [37,60]. In contrast, there are only a few reports on potential virus switching between insects and fungi [53,61]. However, the phylogenetic closeness shared between viruses of sea lice and fungi was revealed by '*Hypoviridae*-related', '*Partitiviridae*-related' and '*Ghabrivirales*-related' viruses, suggesting that virus transmission between arthropods and fungi may have happened numerous times during their evolution. Likewise, we found viruses in sea-lice that belonged to the groups *Endornaviridae*, '*Tombusviridae*-related', and '*Virgaviridae*-related' that were closely related to viruses associated with nematodes, anemones, and algae (Fig 3). These results are consistent with other evidence [37,60,62] that cross-phyla and cross-kingdom transmission of RNA viruses has occurred over evolutionary time between arthropods and other organisms.

Our phylogenetic analysis also uncovered a previously unrecognized route of virus evolution, between parasitic tapeworms and arthropods. Specifically, 'Caligus rogercresseyi toti-like virus 3' and the bunya-like sequences identified in *C. rogercresseyi* and *L. salmonis* were closely related to 'Schistocephalus solidus toti-like virus 2' and 'Schistocephalus solidus bunya-like virus', respectively (Fig 4). *Schistocephalus solidus* is a parasitic tapeworm that infects a cyclopoid copepod, a fish, as well as fish-eating water birds.The 'Schistocephalus solidus toti-like virus 2' is transmitted vertically in *S. solidus*; however, whether the bunya-like virus truly infects *S. solidus* needs to be confirmed, as the virus was only found in the total RNAs of the worm and was not detected in the purified viral fraction [63]. This finding suggests that virus transmissions have occurred between copepods and parasitic tapeworms through their ecological interactions. In addition to tapeworms, arthropods vector a variety of eukaryotic parasites, including protozoa, nematodes, flukes, and roundworms. Our finding of a virus switch between copepods and their parasitic tapeworms evokes the possibility of other potential routes for viral evolution.

We also examined evidence for horizontal virus transmission between sea lice and fish, by searching the fish subset of TSA database (NCBI) for RdRp sequences that were similar to those from the viruses we discovered in sea lice. This revealed known and novel lineages of fish-associated viruses in the groups '*Tombusviridae* related' (Fig 3), '*Ghabrivirales* related', and *Mononegavirales* (Fig 4), but we did not find any known fish viruses, or closely related viral sequences in the transcriptomes of sea lice. Nonetheless, the possibility of host switching between sea lice and fish remains [64], given the blood-feeding behaviour of sea lice, the common appearance of sea lice on farmed salmon, the occurrence of fish and lice viruses within the same evolutionary groups, and the fact that our knowledge about viruses infecting fish [65,66] and sea lice is limited.

## Viruses infecting different species and life stages of sea lice

Our analyses showed that viruses associated with different species of sea lice are often grouped phylogenetically (Figs 2 and 4), even when collected from different species and distant geographical regions. This includes samples of *C. clemensi* and *L. salmonis* collected from British Columbia (Canada) for this study, and by Yasuike et al. [67], respectively, while samples of *C. rogercresseyi* were from Chile [34,68]. The observed phylogenetic grouping suggests that many of the extant viruses harbored by sea lice originated from viruses that infected a common ancestor. Moreover, most of these viruses fall within large clades of AAVs (Figs 2 and 4),

implying that they were inherited from ancestral arthropods at some point during their long-term co-evolution with RNA viruses.

Previous studies of hexartovirus and caligrhaviruses infecting *L. salmonis* showed that the viruses were vertically transmitted and present during the entire life cycle of sea lice [10]. Likewise, most of the viruses we identified in *C. rogercresseyi* were associated with multiple life stages, including planktonic stages where they may disperse with ocean currents over dozens of kilometers [69]. Thus, sea lice are reservoirs of diverse viruses throughout their entire life and may carry the viruses over relatively long distances.

## Novel genome organization and putative genes in viruses of sea lice

Many of the near-complete genomes of RNA viruses that we recovered in our study had atypical structures and encoded putative genes that are highly divergent from those of other described viruses. For instance, members of the *Solemoviridae*, including sobemo-like viruses, have monopartite polycistronic RNA genomes containing 4 to 10 ORFs. These ORFs code for a viral suppressor of RNA silencing, a peptidase bearing polyprotein, a RdRp, and a capsid protein [28]. In contrast, the sobemo-like viruses in sea lice have two genomic segments; the larger segment encompasses two overlapping ORFs, corresponding to a hypothetical protein and a RdRp, while the smaller segment encodes a capsid protein. There was also significant sequence similarity between the hypothetical protein from 'Caligus clemensi sobemo-like virus 1' and a putative peptidase from an unclassified virus (Baird Spence virus) in the *Solemoviridae* that was associated with the Argentine ant, suggesting that these hypothetical proteins may encode peptidases. The identification of similar genome organizations in close relatives ('Wenzhou shrimp virus 9' and Khabarov virus) of sobemo-like viruses from sea lice suggests that these unclassified viruses possess genome structures that are distinct from classified members of the *Solemoviridae*. Intriguingly, we also found that the putative capsid-encoding gene in 'Caligus clemensi sobemo-like virus 2' not only shares significant sequence similarity with sobemo-like viruses (e.g., 'Barns Ness beadlet anemone sobemo-like virus' and 'Wuhan pillworm virus 3'), but also with noda-like (e.g., 'Beihai noda-like virus 11' and 'Rice Noda-like virus') and permutotetra-like (e.g., 'Viola philippica permutotetra like virus' and 'Hubei permutotetra-like virus 11') viruses. The fact that the capsid genes are encoded by subgenomic RNAs or genomic segments in these viruses raises the possibility that horizontal gene transfer may have resulted from mispackaging during co-infection of the same host.

Similarly, novel putative genes were also present in the genomes of the newly identified hypo-like, virga-like, tombus-like, and picorna-like viruses (Fig 7), expanding the known sequence space encompassed by major evolutionary groups of RNA viruses.

## Identification of viral pathogens in sea lice

By comparing the longevity of wild copepods with that of predation-free lab cultured copepods, it was estimated that up to 35 percent of pelagic copepod mortality was non-consumptive mortality [70]. Indeed, previous studies showed that certain RNA viruses can exert catastrophic impacts on the survivals of their crustacean hosts. For example, Taura syndrome virus, a member of the *Picornavirales*, killed 73–87% of penaeid shrimp after viral injections [71]. Leveraging meta-omics techniques, such as characterizing viRNAs [19], quantifying viral replicative strands [72] and identifying ribosome-linked mRNA-rRNA chimeras [73] replicating viruses can be revealed. Yet, it is also important to note that viral infection does not necessarily cause disease, thus may not significantly affect host fitness. To identify viral pathogens of sea lice requires challenge experiments in which fitness is directly measured. Nevertheless, by analyzing transcriptomes and viRNAs, we show that divergent RNA viruses actively infect

wild populations of sea lice, and provide genetic blueprints that can be used to interrogate other crustacean zooplankton to understand the distribution and role of viral infection in sea lice.

### Differences in viral composition among copepod sexes and life stages

Analysis of sequencing libraries from *C. rogercresseyi* (Figs 8 and 9) revealed previously unknown viruses, including relatively abundant picorna-like and toti-like viruses. The relative abundance of different viruses among sea-lice samples also varied; however, although each sample comprised ten sea lice, there were no replicates. Consequently, the variation in viral composition among sea-lice samples cannot be ascribed to differences in sex or age of the sea lice. Nonetheless, the data uncovered numerous previously unknown viruses, and showed large differences in virus composition among pooled sea-lice samples. Further research is needed to determine if such differences are linked to sex or age of the sea lice, methodological differences, or reflect natural variation in the composition of viral assemblages within sea lice populations.

### Conclusion

Our investigations of RNA viruses associated with sea lice have greatly expanded the known viral diversity in crustaceans, in general, and in parasitic copepods, specifically. Moreover, by analyzing small RNAs we demonstrated that many of these viruses replicate in sea lice, that small RNAs are produced in sea lice in response to viral infection, and have presented an approach to recognize viral infections in wild populations of copepods. Finally, through phylogenetic analysis we uncovered previously unrecognized examples of horizontal virus transmission between arthropods and other eukaryotes that have occurred over evolutionary time, and provided compelling examples of the important role that arthropods have played in the evolution of RNA viruses.

## Materials and methods

### Sample collection

Individuals of *C. clemensi* were collected from four species of salmon, i.e., coho (*Oncorhynchus kisutch*), chum (*O. keta*), pink (*O. gorbuscha*) and sockeye (*O. nerka*), as well as from Pacific herring (*Clupea pallasii*). Fish were captured by purse seine at five sampling sites ranging from southern Quadra Island to northern Johnstone Strait, British Columbia, Canada, during summer (May to July) from 2015 to 2017 (Department of Fisheries and Oceans Canada permits: XR422015, XR922016, XR252017). Fish were sampled individually from the net, euthanized with Tricaine methanesulfonate (MS-222) in accordance with Animal Care Protocol A16-0101 of The University of British Columbia, following the Canadian Council on Animal Care guidelines, and immediately frozen in liquid nitrogen. Motile-stage sea lice (i.e., pre-adults and adults) were picked off the fish and based on morphological features observed under a dissecting microscope, categorized by species, sex, and life stage, and immediately frozen at -80°C. Additionally, we collected raw transcriptomic reads from major life stages of *Caligus rogercresseyi* (accession numbers SRR15498757 to SRR15498762).

### RNA extraction and sequencing libraries construction

Total RNA extraction was performed using the kit Direct-zol RNA Miniprep Plus (Zymo Research Corporation). Thirty individuals of *Caligus clemensi* collected from Chum (n = 15) and Sockeye (n = 15) salmon in 2016 were pooled together and homogenized in TRI reagents

(Direct-zol RNA Miniprep Plus) with a pellet pestle, and RNA was extracted from the homogenate following the manufacturer's instructions. The ribosomal RNA of eukaryotes and bacteria were reduced from the total RNA with the Illumina Ribo-Zero Plus rRNA Depletion Kit, and the sequencing library was constructed using the NEBNext Ultra RNA Library Prep Kit. To ensure enough sequencing depth for virus discovery, the sea lice library was evenly mixed with two other copepod libraries and was sequenced on a single lane of the Illumina HiSeq platform (2 x 150 bp PE mode).

## Sequence assembly and viral sequence detection

Raw reads from adults of *Caligus clemensi* (accession number: SAMN17831641) and major life stages of *Caligus rogercresseyi* (accession numbers: SRR15498757 to SRR15498762) were trimmed by Trimmomatic v0.38 [74], and the quality of post-QC reads was checked with FastQC (http://www.bioinformatics.babraham.ac.uk/projects/fastqc/). Eukaryotic and bacterial rRNA were further removed from the post-QC reads using SortMeRNA v4.1 [75]. All the remaining post-QC reads were de novo assembled using both Trinity and rnaSPAdes [76,77]. Contigs generated from both tools were searched in the NCBI-nr database using DIAMOND BLASTx (p value cutoff: 1e-3). To minimize the chance of false positives, only contigs with top hits to viral RdRps were considered as virus-like sequences and retained for further analyses. The authenticity of the virus-like sequences was verified by checking 1) if they were obtained by both assemblers (sequences with 100% identity), 2) if they do not contain any genes that are closely related to those of eukaryotes, EVEs, or transposable elements, and 3) if they do not have exact copies in sea lice genomes (BLASTn).

## Discovery of sea-lice-associated RNA viruses and EVEs in databases

For the identification of viruses from publicly available transcriptomes of sea lice, ORFs from the newly discovered viruses were predicted using ORFfinder (https://www.ncbi.nlm.nih.gov/orffinder/). Notably, these newly discovered viruses were identified through the sequence alignment (BLASTx) against the entire NCBI-nr database as described above. Then, the inferred RdRp coding sequences as well as sequences of other viral genes were further validated by searching against the Conserved Domain Database (CDD) using default settings (https://www.ncbi.nlm.nih.gov/Structure/cdd/wrpsb.cgi). Amino-acid sequences of the CDD-validated RdRps identified in the novel RNA viruses were used as probes to search (tBLASTn) against all the available copepod transcriptomes in the TSA database at NCBI. For the identification of EVEs, virus-like sequences (> 600 nt) were searched against all *Caligidae* protein and nucleotide sequences uploaded to the NCBI databases. Sequences with significant hits (p value < 1e-5) were verified for their authenticity as described above.

## Phylogenetic analyses

Only viral RdRp sequences containing at least 300 aa were selected for further phylogenetic analyses. To resolve evolutionary relationships among novel sea-louse-associated viruses and known RNA viruses, RdRp sequences of representatives from all established viral genera were collated if they were closely related to the newly found viruses in sea lice. To construct phylogenetic trees, RdRp-encoded proteins were first dereplicated by CD-hit [78]. Then, the remaining sequences were aligned for each viral group in MAFFT [79] before being quality screened by TrimAl under the strict mode [80]. The trimmed alignments were manually examined in Jalview [81]. Finally, phylogenetic trees were inferred using the Maximum likelihood method in IQ-TREE 2 [31]. For each sequence alignment, the best amino-acid substitution model was

selected based on Bayesian information criteria (BIC) by ModelFinder [32]. Branch supports were estimated with 1000 ultrafast bootstrap replicates [82].

## Analysis of sRNAs in sea lice

sRNA sequencing data for *C. rogercresseyi* and *L. salmonis* were downloaded from Sequence Read Archive at NCBI, and the fastq files were extracted with 'prefetch' and 'fastq-dump' utilities from the SRA Toolkit, respectively (http://ncbi.github.io/sra-tools/). The accession numbers of the sRNA datasets for each species are as follows: SRR10426891 to SRR10426895 are for sRNA datasets of *C. rogercresseyi* [34], while SRR7749757 and SRR7749758 are for *L. salmonis* [83]. The small-RNA libraries were constructed using TruSeq Small RNA Kit (Illumina, USA) and sequenced on Illumina MiSeq platforms. Details of the methods applied for library preparation, quality control, and high-throughput sequencing have been well-described in the original studies [34,83].

'Trim Galore!' was used to automate quality and adapter trimming, as well as quality test for reads of sRNAs (https://www.bioinformatics.babraham.ac.uk/projects/trim_galore/). Specifically, reads shorter than 17 nt were removed (—length 17), and if the number of adapter-attached sequences was less than 100 in the first one million reads (—consider_already_-trimmed 100), the reads were considered as 'adapter-trimmed' and quality checked directly. For each louse species, the post-QC reads from multiple datasets were merged into a single fasta file. Then, for each species, the post-QC sRNAs were mapped to the corresponding virus-like sequences using Bowtie2 under the sensitive mode [84]. The resulting SAM files were then transformed to BAM files and sorted by Samtools for further analyses [85]. Finally, the size distribution and 5' base composition of sRNAs that mapped to viral sequences were analyzed and plotted using a python script developed by Lewis et al. [86].

## Analysis of taxonomic composition in transcriptomes of *C. clemensi*

The rRNA-depleted, post-QC reads were used to quantify abundance of contigs using algorithms of the bioinformatic tool Salmon v1.3 [87]. TPM (transcripts per million) normalization was performed by Salmon v1.3 to estimate the relative abundance for each contig. Contigs were aligned against the NCBI-nr database using DIAMOND BLASTx (p value cutoff: 1e-3) as described above; we kept records for the top 10 BLAST hits, and retrieved the taxonomy IDs for the reference sequences. R packages 'taxizedb' (https://CRAN.R-project.org/package=taxizedb) and 'tidyverse' [88] were used for obtaining taxonomic information through species taxids and data wrangling, respectively. A contig was classified to the taxon with the greatest number of hits (n = 10). Finally, the TPM for individual contigs were summed for each taxon.

## Profiling of abundances of RNA viruses detected in transcriptomes of *C. rogercresseyi*

For each of the analyzed transcriptomic datasets of *C. rogercresseyi*, TPM normalization of the rRNA-depleted, post-QC reads was performed using Salmon v1.3 [87], as described above. The relative abundance of each type of viruses in each transcriptomic dataset was calculated as: the sum of TPM of viral sequences assigned to a taxonomic group divided by the sum of TPM of all viral sequences detected in the dataset.

## Illustration

Figures were generated using the R package 'ggplot2' [89] and modified as needed using Adobe Illustrator (https://www.adobe.com/).

## Supporting information

**S1 Table. Summary of the phylogeny-resolved RNA viruses found in sea lice.**
(XLSX)

**S1 Fig. The taxonomic composition of the analyzed transcriptomes of *Caligus clemensi*.**
(TIF)

## Acknowledgments

We thank Amy M. Chan, Kevin Zhong, and all members of the Suttle lab, as well as Yanting Liu and Gideon Mordecai for their feedback throughout the project. We greatly appreciate the help of Julian Gan, Carly Janusson, and Brett Johnson as well as other members of the Hakai Institute for providing sea lice samples.

## Author Contributions

**Conceptualization:** Tianyi Chang, Brian P. V. Hunt, Curtis A. Suttle.

**Data curation:** Tianyi Chang.

**Formal analysis:** Tianyi Chang.

**Funding acquisition:** Junya Hirai, Curtis A. Suttle.

**Investigation:** Tianyi Chang.

**Methodology:** Tianyi Chang.

**Resources:** Curtis A. Suttle.

**Software:** Tianyi Chang.

**Supervision:** Curtis A. Suttle.

**Validation:** Tianyi Chang.

**Visualization:** Tianyi Chang.

**Writing – original draft:** Tianyi Chang.

**Writing – review & editing:** Tianyi Chang, Brian P. V. Hunt, Junya Hirai, Curtis A. Suttle.

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
