## [Decision Letter · Decision Letter 0]

15 Dec 2022

Dear Dr. Suttle,

Thank you very much for submitting your manuscript "Pathogens of pathogens: divergent RNA viruses in sea lice" for consideration at PLOS Pathogens. As with all papers reviewed by the journal, your manuscript was reviewed by members of the editorial board and by several independent reviewers. In light of the reviews (below this email), we would like to invite the resubmission of a significantly-revised version that takes into account the reviewers' comments.

Specifically, the Reviewer #3 raised major concerns affecting the title, datasets, analysis, presentation, and conclusions, which require careful consideration. Also a confusion noted by the Reviewer #1 and summary of 32 new viral genomes, asked by the Reviewers #2 and #3, must be addressed.   

We cannot make any decision about publication until we have seen the revised manuscript and your response to the reviewers' comments. Your revised manuscript is also likely to be sent to reviewers for further evaluation.

Sincerely,

Jens H. Kuhn

Academic Editor

PLOS Pathogens

Alexander Gorbalenya

Section Editor

PLOS Pathogens

Kasturi Haldar

Editor-in-Chief

PLOS Pathogens

orcid.org/0000-0001-5065-158X

Michael Malim

Editor-in-Chief

PLOS Pathogens

orcid.org/0000-0002-7699-2064

Reviewer's Responses to Questions

**Part I - Summary**

Reviewer #1: Chang et al. report a detailed survey of potential RNA viruses, identified via an analysis of transcriptomes from sea lice (Caligidae). The analysis included search for both virus-derived RNA sequences and potential virus-interfering RNA molecules. The study was as comprehensive as possible and it dramatically expands the known repertoire of Caligidae-infecting RNA viruses.

The largest limitation, besides being limited by the data, available to this date, is using the NCBI CDD database for identification of RNA virus genes, RdRp in particular. The collection of profiles, available through CDD at this point, is way behind the state of the art, especially considering the recent expansion of the field, driven by large-scale metatranscriptomic studies. Some of the more exotic RdRps are poorly recognized by the CDD profiles and could have been missed. On the other hand, using the whole-CDD search (rather than a targeted search with an RdRp profile collection) provides a good protection against false positives (misidentifying DdRps and RTs as RdRps), so the author's choice could be considered as a reasonable compromise ("safe" over "bold").

The results, presented by the authors, contain a surprise that gives me an uneasy feeling, the dramatic difference between the virus repertoires, discovered in Caligus rogercresseyi and Caligus clemens (Fig. 3, compare the sharp contrast between C. rogercresseyi and C. clemens to the gradual change over the life stages of C. rogercresseyi). Thee is no hard reason why their viromes could NOT differ this much, so this is not necessarily indicative of an error. Still, if possible at all, I would suggest that the authors take another look at this; maybe it is explainable by some difference in sample preparation or processing?

Reviewer #2: The authors have discovered 32 new viral genomes using transcriptomic data from the species Caligus rogercresseyi, Caligus clemensi and Lepeophtheirus salmonis. For C. rogercresseyi and L. salmonis they used publicly available data, while the transcriptome of C. clemensi was sequenced by the authors. The authors expand the caligid virosphere extensively as only 5 viruses were previously known. The analysis of viRNA in the caligid host was a clever way to investigate if the viruses were actively replicating in the caligid.

Methodology is thorough and clear. However, I do think the article could benefit from a clearer presentation of the 32 new viral genomes.

Reviewer #3: The study by Chang et al reveals the diversity of RNA viruses in sea lice, an economically important group of arthropod parasites that infest fish. Only three species of sea lice were examined here, including Caligus clemensi for which the authors carry out total RNA sequencing, and Caligus rogercresseyi and Lepeophtheirus salmonis for which they analyzed the sequencing data obtained from SRA database. A total of 32 new virus species were identified, although none of them are likely to be associated with diseases/infections in fish (i.e. vector-borne virus). Based on these discoveries, the authors describe their diversity, evolutionary history, and abundance levels across different sea lice species, life stages and sex. Furthermore, they use the presence of canonical virus-derived small interfering RNAs as evidence to support the claim that these viruses, including those related to fungus or other basal eukaryotes, are associated with the principal host (i.e. see lice).

Overall, the topic is of great interest, but the study is descriptive in nature, has very limited sample size, and is flawed in study design. Furthermore, I would be very cautious and avoid general conclusions such as “cross-phylum and cross-kingdom switching between arthropods and a wide array of eukaryotes” (lines 42-45), “extensive transmission of viruses between arthropods and other eukaryotes” (lines 58-59) and “viruses that may affect sea-lice survival” (line 46), since the evidence present in this study are far from enough for such statement.

**Part II – Major Issues: Key Experiments Required for Acceptance**

Reviewer #1: No major issues detected.

Reviewer #2: (No Response)

Reviewer #3: 1. If this study aims to explore the RNA virus diversity in sea lice, the current sampling and sequencing is massively inadequate for such task. I would expect multiple sequencing runs representing different species and different populations of sea lice that are sampled from multiple geographic locations. However, only three species were involved in this study (lines 481-494), and among which only one (i.e. C. clemensi) is sampled and processed by the authors (30 individual in a single pool, lines 498). The rest of the sequencing data (C. rogercresseyi and L. salmonis) are downloaded from the database, and these data are limited as well. Therefore, I would expect more sequencing data from the authors to cover more species or populations of sea lice. Or, at the very least, to download more SRA data associated with sea lice. Indeed, more SRA reads are available for sea lice than those linked to TSA database.

2. The authors intend to compare virome between different life stages, gender, and species. However, there is no experimental or sequencing design for these comparisons such that I am not convinced of the results and conclusions made here (lines 132-146). First, there is no replications associated with each treatment group, and as a result no statistical significance can be derived from any of the comparisons made here. For example, in the case of C. rogercresseyi, each life-stage is represented by a single library, and therefore the variations observed here (Fig. 2) are very likely to be subject to random effect. Furthermore, comparisons are made across libraries that are derived from very different sampling and sequencing procedures (Fig. 3) which involve entirely different studies. And any of the following can have huge impact on the results: for example, whether they use the same sampling processing and storage conditions? And the same procedures for RNA extraction and library preparation? Which sequencing platforms are used? And how many individuals are mixed in the same pool, amongst others.

3. The authors use virus-derived small RNAs (viRNA) as evidence of “active replications” of viruses within the principal host (i.e. sea lice) and claim that even fungal and plant-like viruses can infect sea lice and are likely to be “disease agents” (lines 343-344), and that they are derived from numerous “horizontal virus transmission between arthropods and fungi, nematodes, anemones, and algae” (lines 372-379). Such strong conclusions could not be made based on viRNA profiling alone. My impression, based on authors results, is that ALL RNA viruses (except for EVEs) reported here present canonical viRNAs similar to those found in other arthropods (lines 218-219). Nevertheless, I did not see viruses with a different small RNA profile such that it could be assigned to a different host group other than sea lice (i.e. parasites, symbionts, food, contaminations…)? If every single non-EVE virus is infecting sea lice, how can the authors make a distinguishment? It will be useful if the authors can provide small RNA profile of viruses that are not associated with the principal host. Otherwise, the author should consider the following scenario: (i) viruses are targeted by arthropod anti-viral system simply due to its presence within arthropod but not “actively replication”, (ii) the small RNA profile of plants or basal eukaryotes might be similar to that of arthropods, which have already been reported elsewhere.

**Part III – Minor Issues: Editorial and Data Presentation Modifications**

Reviewer #1: The nature of the data in Figure 3 just begs for an ordination-style plot (projecting a multidimensional dataset into a low-dimensional space, e.g. a plane). This would make a nice visualization.

Reviewer #2: It is mentioned that they have discovered 32 new viruses, but it is not easy for the reader to get a clear overview of what they have discovered. I think they are all represented in the phylogenies, and some are presented in Figure 9. But I think It would be beneficial to sort all new viral sequences in a list showing short description, sequence length, partial/complete, actively replicating based on viRNA (yes/no/not investigated) and accession number.

Reviewer #3: 1. Title: “pathogens of the pathogens”, there is no evidence that these viruses are pathogens, even assuming they are actively replicating within sea lice. It could be symbiotic organisms.

2. The authors should provide a table or supplementary table which presents details of each viruses discovered in this study, such as virus names, host species, sequence length, taxonomy lineage, blast hits, abundance levels, amongst others.

3. Fig 1. The authors should not use a schematic phylogeny to show the diversity or taxonomy groups of the newly discovered viruses. The phylogeny is not derived from the author’s data and is subject to high controversy.

4. Line 133 and fig 2. TPM is not addable across different transcripts and should not be used as an estimate of total abundance. The authors should use RPM or other statistics instead.

5. Line 142. It is unclear what “relative abundance” represents and how it is estimated.

6. Line 203. “deeply branching lineage”: poor phrasing

7. Lines 389-390. “ancient virus transmission between …”. It is highly speculative statement, suggest deletion.

8. Lines 397-401. This is part of the results, please move this section from Discussion to Results

9. Lines 442-443. “…novel structures…” bad phrasing

10. There is no effort to discuss the potential pathogenicity of viruses discovered

11. Lines 536-527. The authors should not use conserved domain database as the only database for RdRp identification, because its RdRp collection is far from comprehensive.

12. Lines 542. Why “identical sequences were merged but marked as different viruses”?

13. All samples were sequenced in a single pool here. Although individual specimens are morphologically identified, are there any analyses of host marker gene (e.g. COI) to confirm that they belonged to the same species.

PLOS authors have the option to publish the peer review history of their article (what does this mean?). If published, this will include your full peer review and any attached files.

Reviewer #1: No

Reviewer #2: No

Reviewer #3: No
---

## [Editor Report · Decision Letter 1]

9 Mar 2023

Dear Dr. Suttle,

Thank you very much for submitting your manuscript "Divergent RNA viruses infecting sea lice" for consideration at PLOS Pathogens. As with all papers reviewed by the journal, your manuscript was reviewed by members of the editorial board and by several independent reviewers. The reviewers appreciated the attention to an important topic. Based on the reviews, we are likely to accept this manuscript for publication, providing that you make some modifications:

1. Please retain the data on sex and age but with a caveat statement that they were not analyzed in this study but provided for future studies with increased sampling. It would be a disservice to the community not to make collected data available.

2. Please change the title to “Divergent RNA viruses infecting parasitic sea lice” or “Divergent RNA viruses infecting sea lice, major ectoparasites of fish” to make this aspect explicit for the broad readership without overinterpreting the available data.

3. All virus taxon names, for instance those ending in “-rnaviricota” in Fig. 1, should be italicized (whereas virus names should not be).

Sincerely,

Jens H. Kuhn

Academic Editor

PLOS Pathogens

Alexander Gorbalenya

Section Editor

PLOS Pathogens

Kasturi Haldar

Editor-in-Chief

PLOS Pathogens

orcid.org/0000-0001-5065-158X

Michael Malim

Editor-in-Chief

PLOS Pathogens

orcid.org/0000-0002-7699-2064

"Divergent RNA viruses infecting sea lice" is the article title I would prefer based on reviewer's concerns.

Reviewer Comments (if any, and for reference):

Figure Files:

Data Requirements:

Reproducibility:

References:

---

## [Editor Report · Decision Letter 2]

25 Apr 2023

Dear Dr. Suttle,

We are pleased to inform you that your manuscript 'Divergent RNA viruses infecting sea lice, major ectoparasites of fish' has been provisionally accepted for publication in PLOS Pathogens.

Best regards,

Jens H. Kuhn

Academic Editor

PLOS Pathogens

Alexander Gorbalenya

Section Editor

PLOS Pathogens

Kasturi Haldar

Editor-in-Chief

PLOS Pathogens

orcid.org/0000-0001-5065-158X

Michael Malim

Editor-in-Chief

PLOS Pathogens

orcid.org/0000-0002-7699-2064
---

## [Editor Report · Acceptance letter]

30 May 2023

Dear Dr. Suttle,

We are delighted to inform you that your manuscript, "Divergent RNA viruses infecting sea lice, major ectoparasites of fish," has been formally accepted for publication in PLOS Pathogens.

Best regards,

Kasturi Haldar

Editor-in-Chief

PLOS Pathogens

orcid.org/0000-0001-5065-158X

Michael Malim

Editor-in-Chief

PLOS Pathogens

orcid.org/0000-0002-7699-2064